# Not All That Glitters Is Gold: Barcoding Effort Reveals Taxonomic Incongruences in Iconic Ross Sea Sea Stars

Alice Guzzi [1,2,*], Maria Chiara Alvaro [2], Bruno Danis [3], Camille Moreau [3] and Stefano Schiaparelli [2,4]

1 Department of Physical Sciences, Earth and Environment (DSFTA), University of Siena, 53100 Siena, Italy
2 Italian National Antarctic Museum (MNA, Section of Genoa), University of Genoa, Viale Benedetto XV No. 5, 16132 Genoa, Italy; chiara.alvaro@yahoo.it (M.C.A.); stefano.schiaparelli@unige.it (S.S.)
3 Marine Biology Lab, Université Libre de Bruxelles (ULB), B-1050 Brussels, Belgium; bruno.danis@ulb.be (B.D.); mr.moreau.camille@gmail.com (C.M.)
4 Department of Earth, Environmental and Life Sciences (DISTAV), University of Genoa, Corso Europa 26, 16132 Genoa, Italy
* Correspondence: aliceguzzi@libero.it

**Abstract:** The Southern Ocean is one of the most exposed regions to climate-related changes on our planet. Better understanding of the current biodiversity and past speciation events, as well as implementation of conservation actions and accurate identification of organisms to species level in this unique environment, is fundamental. In this study, two species of sea stars, *Odontaster roseus* Janosik & Halanych, 2010 and *Odontaster pearsei* Janosik & Halanych, 2010, are reported for the first time from the Terra Nova Bay area (TNB, Ross Sea, Antarctica) by using a combination of molecular (DNA barcoding) and morphological (coloration and skeletal features) analyses. Molecular results agree with external morphological characters of the two identified species, making occurrence in the area unequivocal. The two species were recently described from the Antarctic Peninsula, and went unnoticed for a long time in TNB, possibly having been confused with *O. meridionalis* (E.A. Smith, 1876), with which they share a bright yellow coloration. This latter species seems to be absent in the Ross Sea. Thus, the past literature referring to *O. meridionalis* in the Ross Sea should be treated with caution as these "yellow morphs" could be one of the two recently described species or even orange–yellow morphs of the red-colored congeneric *O. validus* Koehler, 1906. This work highlights the paucity of knowledge even in purportedly well-studied areas and in iconic Antarctic organisms.

**Keywords:** Southern Ocean; COI; morphology; *Odontaster*; Asteroidea



## 1. Introduction

Asteroidea (sea stars) is one of five extant classes belonging to the phylum Echinodermata. The class includes 38 families and approximately 1900 species [1–3], making it the second most diverse echinoderm class after the Ophiuroidea [4–6]. Sea stars show high ecological diversity and are important components of marine ecosystems where they occur, from the intertidal to hadal depths (9990 m) [7–9]. In the Southern Ocean (SO), asteroids are well represented, accounting for 15% to 16% of the total number of species reported there to date [1,10,11]. Current diversity estimates for this class south of 45° S count 28 asteroid families, 118 genera, and 299 species [12]. As with other invertebrates thriving in polar environments, Antarctic sea stars have developed specific adaptations (e.g., slow development [13,14]) and reproductive strategies (brooders vs. broadcasters [12,15]) that affect distribution patterns and the biogeography of this class in the SO [16].

Although many species of sea stars can be identified based on morphological characteristics, their phenotypic diversity at the species level is commonly so high that taxonomic boundaries can be challenging (or even impossible) to morphologically determine [17–19].

With the rapid accumulation of samples in museums and the co-occurring decline of taxonomic expertise in recent years [20], cladistics, phylogenetics, and coalescent-based analyses have become key tools for species identification or discrimination.

Although some evolutionary relationships between asteroid families and species are still to be conclusively assessed, the implementation of molecular tools and the availability of data during the last 20 years have allowed a great leap in accuracy of knowledge for this taxon (e.g., [21–26]).

Molecular tools differ in effectiveness and interpretation in relation both to the research question and the unique evolutionary histories of the taxa [27]. They are proving particularly useful and efficient in the case of Antarctic sea stars. Indeed, the isolation of the Antarctic continent (which started in the Oligocene) resulted in typically high levels of endemicity in the SO shelf fauna [28–30].

Use of DNA barcoding has increased since its introduction in 2003 as a routine tool for species identification, to effectively discriminate species and "unmask" those that look similar. In particular, the barcode gap, thanks to interspecific genetic variation being generally higher than intraspecific ones, often allows correct delineation of species [31]. An integrative approach to taxonomy, i.e., by using morphological characteristics as well as one to several genes, is necessary for assessing species richness and species boundaries in many or most situations [32].

Few molecular studies have been performed on SO asteroids, and they have focused on the abundant, near-shore genus *Odontaster* (e.g., [33–37]), making it one of the most studied echinoderms in Antarctica.

This genus occupies a key trophic position in shallow benthic communities of the Southern Ocean [38,39]. *Odontaster validus* Koehler, 1906, in particular, has been used as a model species in studies in Antarctic water focusing on distribution and abundance (e.g., [40,41]), metabolism (e.g., [36]), ocean acidification (e.g., [42]), isotopic trophic position (e.g., [43]), and consequences of physical climate change on Antarctic organisms (e.g., [44,45]).

Despite the numerous scientific publications on this model genus, recent updates on *Odontaster* taxonomy [34,35] highlighted that its diversity might be higher than recorded, even in well-studied areas.

Two species within the *Odontaster* genus were fairly recently described from the Antarctic Peninsula region, *O. roseus* Janosik & Halanych, 2010 and *O. pearsei* Janosik & Halanych, 2010, and set out the problem of redundant errors due to lack of resources for identification and consistent taxonomic revision. Specifically, these two species should not have to be considered a cryptic species (which display no obvious morphological differences) but are referred to as "unrecognized biodiversity" having clear diagnostic morphological characters (e.g., the number of spines on abactinal plates, spine length, as well as differences in marginal plates and marginal spines) that has escaped previous detection [35]. This pattern of unrecognized species diversity is common in the SO (e.g., [46–49]) and many authors have highlighted the efficiency of integrated molecular and morphological techniques as a fundamental explorative tool to unravel marine biodiversity (e.g., [50,51]).

The Ross Sea area is one of the most productive regions in the Southern Ocean [52]; and since December 2017, it has fallen under the protection of the Conservation Measure 91-05 (2016), which declared it a Marine Protected Area (RSRMPA). Nevertheless, a specific assessment of the molecular diversity of sea stars has never been performed.

Since 1985, the Italian National Antarctic Research Program (PRNA) has coordinated several research activities and gathered extensive biological and oceanographic information, resulting in a rich specimen collection.

In this framework, sea stars were targeted by several studies (e.g., [53,54]), while a first complete faunistic inventory of asteroids from the Terra Nova Bay (hereafter TNB) area (30–500 m depth) was published by Chiantore et al. [55]. Chiantore et al. [55] identified 15 different sea stars species belonging to seven families, with genus *Odontaster* comprising two species, i.e., *Odontaster validus* Koehler, 1906 and *Odontaster meridionalis*

(E. A. Smith, 1876). These two taxa were discriminated by morphological traits, mainly relying on Clark [56]. The Asteroid check list for TNB has not been updated since then. The same two taxa were repeatedly cited in other studies performed in the Ross Sea, especially in the McMurdo area (e.g., [14,57,58]).

*Odontaster* species are «model species» in a variety of field studies as well as benthic monitoring programs for the Ross Sea region Marine Protected Area by the RSMPA monitoring plan (CCAMLR Conservation Measure 91-05: Ross Sea Region Marine Protected Area. 2016 [59]). The possible presence of unnoticed diversity in the genus *Odontaster* led us to re-evaluate the biodiversity of this genus for the TNB area. Hence, the objective of our study was to perform molecular and morphological analysis on *Odontaster* samples collected by Italian National Antarctic Program (PNRA) and curated by the Italian National Antarctic Museum (MNA, Genoa section).

## 2. Materials and Methods

The study area is Terra Nova Bay, which is commonly ice-free during polar summer months. The region is located on the western margin of the Ross Sea and stretches from Cape Washington Peninsula (74°44′ S 163°45′ E), in the north, to the floating tongue of the Drygalski glacier (64°43′ S 60°44′ W), arising from David Glacier in the south [60] (Figure 1). The Terra Nova Bay polynya (TNBP), an open water area surrounded by sea ice [61], is a part of both the marine protected area and the Antarctic Special Protected Area (n.161) in the western Ross Sea [62]. The bay comprises a tortuous continental shelf with numerous banks and deep embayments. The mean depth of the shelf is approximately 450 m, with the greatest depths close to the coast and areas up to 1000 m deep in the adjacent basin.

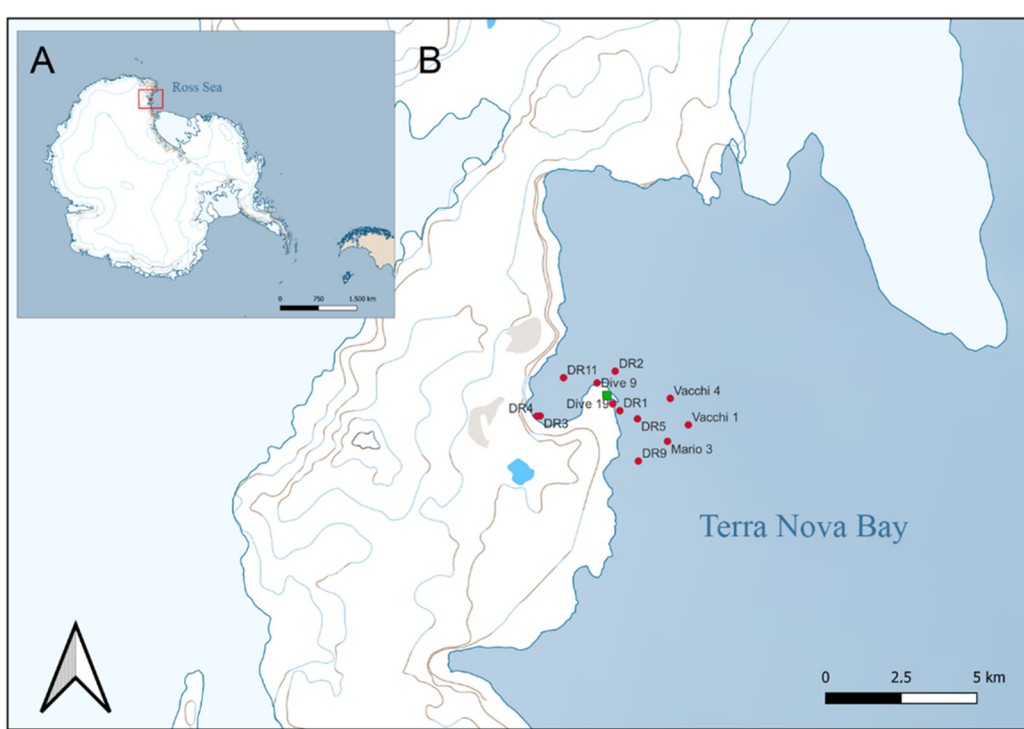

**Figure 1.** Antarctica (**A**) with detail of Terra Nova Bay (Ross Sea) and (**B**) sampling sites with Mario Zucchelli Station (Italy) highlighted in green.

Data presented here were collected in the framework of four different Italian PNRA research projects:

- 2006/08.01 ("The coastal ecosystem of Terra Nova Bay" in the Latitudinal Gradient Program—LGP) ("XXV" expedition, 2009/2010).
- 2010/A1.10 (BAMBi; Barcoding of Antarctic Marine Biodiversity) ("XXVII" expedition, 2011/2012 and ("XXVIII" expedition, 2012/2013).

-　　2009/A1.09 (Diversità genetica spazio temporale di endoparassiti delle regioni polari: uno studio per la valutazione dell'impatto dei cambiamenti globali sulle reti trofiche marine) ("XXVIII" expedition, 2012/2013).

### 2.1. Sampling and DNA Extraction

A total of 40 samples belonging to the *Odontaster* genus were analyzed and the distributional data considered here originated from 13 different sampling stations, ranging between 15 and 569 m of depth (Table 1). Sampling was performed through deployments of a variety of sampling gear. Benthic sampling under the Italian PNRA was mainly performed using a rectangular dredge (70 × 30 cm) and an unconventional set of gears for sampling benthic fauna (such as a trammel net and a small Hamburg plankton net) that opportunistically collected benthic specimens due to accidental contact with the bottom during gear deployment "failures". Two samples were photographed and collected by Stefano Schiaparelli during SCUBA diving, performed in the framework of the PNRA "XXV" Expedition (2009/10) along the rocky cliffs of Tethys Bay "Zecca" and Road Bay.

**Table 1.** Sampling stations and data. Abbreviations: Mario Zucchelli Station (MZS); number of specimens (*n*).

| Expedition | Station | Location | Year | Latitude | Longitude | Depth (m) | Sample Vouchers | N |
|---|---|---|---|---|---|---|---|---|
| PNRA XXV Exp 09/10 | Dive 9 | Tethys Bay "zecca" | 2009 | −74.9027 | 164.10255 | 23 | MNA-02814 | 1 |
| | Dive 19 | Road Bay | 2010 | −74.69647 | 164.12007 | 15 | MNA-02902 | 1 |
| PNRA XXVII Exp 11/12 | DR1 | Road Bay | 2012 | −74.69848 | 164.12812 | 100 | MNA-03430, 04282, 04283 | 3 |
| | DR3 | Tethys Bay | 2012 | −74.70005 | 164.03873 | 60 | MNA-03582 | 1 |
| | DR4 | Tethys Bay | 2012 | −74.70010 | 164.03502 | 198 | MNA-04276 | 1 |
| | DR9 | Faraglione | 2012 | −74.71337 | 164.14903 | 150 | MNA-03791, 03812, 03825, 03832, 03841, 08034, 08035, 08036 | 8 |
| PNRA XXVIII Exp 12/13 | DR5 | Road Bay | 2013 | −74.70087 | 164.14793 | 150 | MNA-05817, 08037, 08038, 08039 | 4 |
| | DR9 | MZS ('fossa') | 2013 | −74.68090 | 164.21433 | 522 | MNA-06116 | 1 |
| | DR11 | Tethys Bay | 2013 | −74.68872 | 164.06493 | 222 | MNA-06486, 06489, 06490 | 3 |
| | Vacchi 1 | Tethys Bay | 2013 | −74.70262 | 164.20502 | 569 | MNA-05430 | 1 |
| | Vacchi 4 | Tethys Bay | 2013 | −74.69478 | 164.18458 | 454 | MNA-06331, 08021, 08022, 08023, 08024, 08025, 08026, 08027, 08028, 08029, 08030, 08031, 08032, 08033 | 14 |
| PNRA XXIX Exp 13/14 | DR2 | "Dorsale" MZS | 2014 | −74.68677 | 164.12278 | 94 | MNA-08043 | 1 |
| | Mario 3 | Punta Stocchino | 2014 | −74.70750 | 164.18167 | 281 | MNA-08042 | 1 |

After collection, samples were transferred to the laboratory, and the significant morphological characteristics of the live specimens were photographed to preserve information about the original coloration of the organisms. After that, samples were stored in ethanol (75% Et-OH) or frozen (−20 °C) for subsequent molecular analysis. Thereafter, samples were acquired by the MNA and included in their collections (available online at https://steu.shinyapps.io/MNA-generale/, accessed on 10 January 2022). All specimens were classified to the lowest possible taxonomical resolution [63] on a morphological basis by using the available literature and keys from Fisher (1940) [64] and Clark (1963) [56].

Stefano Schiaparelli, Alice Guzzi, Bruno Danis, and Camille Moreau contributed to morphological identification of specimens.

For molecular analyses, a portion of tube feet or arm tip tissue was clipped from each sample for DNA extraction and sequencing of partial cytochrome c oxidase subunit 1 (CO1). The molecular analyses were carried out at the Canadian Centre for DNA Barcoding (University of Guelph, Guelph, ON, Canada). Sequences were uploaded to the BOLD platform (Barcode Of Life Data systems, http://www.boldsystems.org, accessed on 8 February 2022). Primers used for amplification were LCOech1aF1 or LCO1490 and HCO2198 (Table 2).

**Table 2.** List of used primers for cytochrome oxidase I (COI) amplification in our work. Forward primers (F) and reverse primer (R).

| Region | Direction | Primer | Sequence (5'-3') | Reference |
|---|---|---|---|---|
| COI | F | LCOech1aF1 LCO1490 | TTTTTTCTACTAAACACAAGGATATTGG GGTCAACAAATCATAAAGATATTGG | Corstorphine, 2010 [65] Folmer et al., 1994 [66] |
| | R | HCO2198 | TAAACTTCAGGGTGACCAAAAAATCA | Folmer et al., 1994 [66] |

Taxonomic assignation was performed manually in the Barcode of Life database (BOLD) and National Center for Biotechnology Information (NCBI) database BLAST (https://blast.ncbi.nlm.nih.gov/Blast.cgi, accessed on 8 February 2022) for definitive assignment. A sequence match of >98% to the reference database was considered an "exact" match [67]. Accepted taxonomic names and classification were obtained from the World Register of Marine Species (WoRMS www.marinespecies.org/, last search 8 February 2022).

Chromatograms were edited in CodonCode Aligner v9.0.1 (CodonCode Corporation, Centerville, Massachusetts, USA; http://www.codoncode.com/aligner/, accessed on 8 February 2022), primers were trimmed, and the absence of stop codon in the sequences was checked with the same software. Sequences were aligned using MUSCLE, available within CodonCode Aligner, and checked by eye. Based on current understanding of sea star relationships [68], *Acodontaster conspicuus* (Koehler, 1920) (accession number: DQ380237) was chosen as the outgroup. The model with the lowest BIC scores (Bayesian information criterion) in MEGA X [69] analysis resulted T92 + G (Tamura 3-parameter + Gamma distribution) and is considered to best describe the substitution pattern. The evolutionary history was inferred in MEGA X using the maximum likelihood (ML) method based on the Tamura 3-parameter model [70]. For completeness, a maximum parsimony (MP) tree was also produced in the software. A Bayesian phylogeny was subsequently produced using Mr Bayes [71,72]. Based on the notion that nonparametric bootstrap frequencies for ML estimates and Bayesian posterior probabilities for clades in phylogenetic trees are not universally equivalent [73] and the possibility of obtaining wrongly supported results with under parametrization in Bayesian inference, the generalized time reversible (GTR) model with gamma(G)-correction was used. Posterior probabilities were calculated by two independent analyses (one cold and three heated chains) using Markov chain Monte Carlo (MCMC) algorithm. Samples of trees and parameters were extracted every 100 steps from a total of $2 \times 10^8$ MCMC generations. The first 25% of trees were discarded as the burning and the remaining were used to interfere a consensus tree. Tracer v.1.6 was used to ensure an appropriate effective sampling size (ESS all > 100). All obtained trees were imported and compared in FigTree v1.4.4 (http://tree.bio.ed.ac.uk/software/figtree/, accessed on 8 February 2022) for graphic implementation. All sequences were deposited in GenBank (accession numbers: MK811555, MK811610, ON103472-ON103509).

### 2.2. Species Delimitation Methods

Throughout our analyses, a phylogenetic species concept, based on the principle that genetic variation between species (interspecific) is greater than the genetic variation within species (intraspecific) [74], was used. Thus, where two or more species are distinct, there should be a lack of overlap between intraspecific and interspecific sequence

variation, commonly referred to as the "barcode gap" [75]. To identify the number of molecular operational taxonomic units (MOTUs) within our dataset, we applied four different methods of species delimitation to propose primary species hypotheses. Two were distance-based: (i) Barcode Index Number (BIN) system [76], (ii) Automatic Barcode Gap Discovery (ABGD) [32] (bioinfo.mnhn.fr/abi/public/abgd); and two were tree-based: (iii) Generalized Mixed Yule Coalescent method (GMYC) [77] (species.h-its.org/gmyc), performed using the single threshold method, and (iv) Bayesian Poisson tree process (bPTP) [78] (species.h-its.org/ptp).

All sequences were barcode-compliant (*n* = 40). They received a Barcode Index Number (BIN), which aided species delimitation [76]. The Automatic Barcode Gap Discovery method (ABGD) is an automatic procedure that considers the sequences as hypothetical species based on the barcoding gap. The model employs a two-phase system, which initially divides sequences into operational taxonomic units (OTUs) based on a statistically inferred barcode gap (i.e., initial partitioning), and subsequently conducts a second round of splitting (i.e., recursive partitioning). The default values of 0.001 to 0.1 were explored as intraspecific distances and in ABGD, gap values from 1 to 1.5 were applied. The ABGD analysis (bioinfo.mnhn.fr/abi/public/abgd) was performed with a relative gap width of one and Kimura (K80) as the genetic distance.

GMYC requires a fully resolved ultrametric tree as input. The tree-based methods employ a coalescent framework to independently identify evolving lineages without gene flow, each representing a putative species [79]. They can be performed using a single marker and are used to establish a threshold that identifies the separation of intraspecific population substructure from interspecific divergence, and therefore identifies those groups that may be candidate species [80]. The last species delimitation approach was implemented using a Poisson tree process (PTP), which models the speciation using the number of substitutions to infer putative species boundaries on a given phylogenetic input tree [78]. It assumes that the number of substitutions between species is significantly higher than the number of substitutions within species [78]. Here, we used the Bayesian implementation of the Poisson tree processes model (bPTP) [78], which uses a phylogenetic tree and is based on the phylogenetic species concept. The ML tree was used as input. The bPTP analysis (species.h-its.org/ptp) was applied using 500,000 generations of Markov chain Monte Carlo, a thinning of 100, and a burn-in of 25%. The outgroup (*Acodontaster conspicuus*) was removed in all delimitation analysis.

### 2.3. Molecular Data Gathering

To add resolution to our analysis we searched the GenBank and BOLD public sequence database records of *Odontaster* COI sequences from the Ross Sea area to perform a review on all existing classified specimens. The BOLD database regularly synchronizes with GenBank, and there is significant duplication with GenBank records. These duplicated records contain GenBank Accession Numbers, which were checked against the GenBank downloaded entries and removed or added as necessary. GenBank records were given priority over BOLD records because, according to the BOLD handbook (https://v3.boldsystems.org/index.php/resources/handbook, accessed on 8 February 2022), all BOLD records are eventually submitted to GenBank. Any records unique to BOLD should therefore eventually be included in GenBank and would then be removed as duplicates. After downloading respective GenBank and BOLD data, duplicated records from BOLD and GenBank were resolved (keeping the GenBank version in cases of duplication). We also decided to include the COI sequences from Janosik et al. [34,35] (GenBank accession numbers: GQ294339-GQ294396) to ensure we had enough representative sequences from each *Odontaster* species identified. All the data retrieved were combined and we ran the molecular analyses with the same settings.

*2.4. Literature Review*

We searched the published scientific literature using two techniques: (i) searches in online databases (Wiley Interscience, Sciencedirect and ISI Web of Knowledge, last search 22 February 2022), and (ii) manual searches in specific journals.

For the first technique we searched each database using the terms combinations: '*Odontaster*' AND 'Ross Sea' and '*Odontaster*' AND 'Terra Nova Bay'. We searched for these terms under 'full text/abstract' in Wiley Interscience, 'abstract, title, keywords' in ScienceDirect and 'topic' in ISI Web of Knowledge, which includes title, abstract, author keywords, and keywords plus®. In the second technique, we conducted searches using online journal home pages (PlosOne, Antarctic Science, Polar Biology, Marine Ecology Progress Series, Nature, Marine Biology, Deep Sea Research, Frontiers in Marine Science, Hydrobiology and Ross Sea Ecology). The papers we identified through this literature search were included for subsequent analyses, but only if they were peer-reviewed and reported on actual *Odontaster* samples from our study area. Therefore, studies documenting other organisms and comparing them to *Odontaster* from Ross Sea or TNB were not included in our study.

## 3. Results

A total of 40 specimens were analyzed in the current study and all were correctly sequenced to obtain a final COI sequence length of 628 bp. The COI dataset employed for analyses is reported as Supplementary Material (M1). Of the 40 sequences generated in this study, 17 belonged to *Odontaster roseus*, 16 to *Odontaster validus*, and 7 to *Odontaster pearsei* (Supplementary File S1). The maximum likelihood and Bayesian analysis results are consistent and reveal three distinct groups corresponding to recognized species of *Odontaster* (Figure 2). Clade I (posterior probability 94.7% ML and value of 1.00 in Bayesian) comprised individuals of *O. roseus*, Clade II (posterior probability 99% ML and value of 1.00 in Bayesian) comprised *O. pearsei* individuals, whereas *O. validus* individuals were included in Clade III (posterior probability 99.9% ML and value of 1.00 in Bayesian). In our samples, no corresponding sequence matched *O. meridionalis*, a species previously reported from Terra Nova Bay water [55].

*3.1. Species Delimitation Methods*

All sequences were barcode-compliant (Table 3) and received a barcode index number (BIN), which aided species delimitation [76]. The other species delimitation methods recovered the same number of secondary species hypotheses, or SSH (Figure 2, Supplementary File S2) three SSH in the total dataset when using ABGD; three SSH using GMYC, and three SSH using bPTP.

**Table 3.** Samples species partition and associated BOLD BIN. Abbreviations: barcode index number (BIN); number of samples (*n*).

| BOLD BIN | Species | *n* | Sample Vouchers |
|----------|---------|-----|-----------------|
| AAE2388 | *Odontaster roseus* Janosik & Halanych, 2010 | 17 | MNA-02814, 03791, 03812, 03832, 03841, 05817, 06486, 06490, 08024, 08025, 08027, 08028, 08033, 08036, 08037, 08038, 08043 |
| AAK3286 | *Odontaster validus* Koehler, 1906 | 16 | MNA-02902, 03430, 03582, 03825, 04276, 04282, 04283, 05430, 06331, 08021, 08022, 08026, 08029, 08030, 08031, 08035 |
| AAO2072 | *Odontaster pearsei* Janosik & Halanych, 2010 | 7 | MNA-06116, 06489, 08023, 08032, 08034, 08039, 08042 |

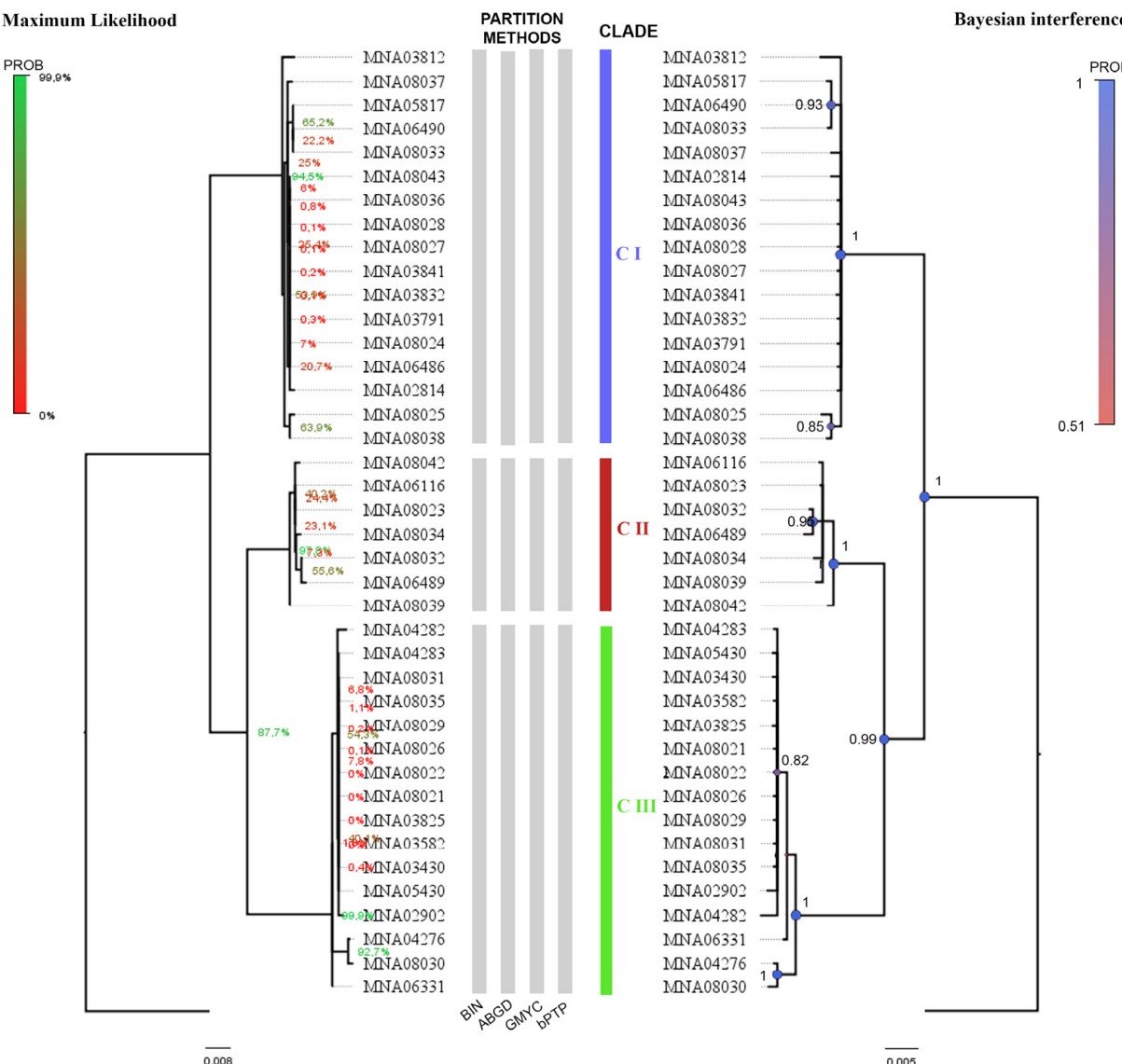

**Figure 2.** Tree topology comparison of maximum likelihood (**left**) and Bayesian interference (**right**). Posterior probability node values are showed on the tree with corresponding legend for each analysis. BIN: barcode index number; BOLD: automatic species delimitation [76]; ABGD: results from automatic barcode gap discovery method [32]; GMYC: species delimitation from generalized mixed Yule coalescent method [77]; bPTP: species delimitation using Bayesian Poisson tree processes method [78]. Clade I (C I) in the figure corresponds to *O. roseus* Janosik & Halanych, 2010; Clade II (C II) corresponds to *O. pearsei* Janosik & Halanych, 2010; and Clade III (C III) to *O. validus* Koehler, 1906.

*3.2. Sequences Database Review*

A total of 105 COI sequences (65 obtained from online data repository and 40 obtained by the current work) were combined in a single dataset and analyzed. Tree topology was inferred using maximum likelihood and Bayesian inference (ML tree available in Figure 3). Species delimitation methods highlighted seven different clades, corresponding to *O. validus*, *O. roseus* (Clade I and II), *O. pearsei*, *O. penicillatus*, and *O. meridionalis* (Clades V and VI, we kept the Janosik et al. [35] nomenclature in Supplementary File S3). In Janosik et al. [35], Figure 3, GQ294370.1 (Sample ID "As 60") corresponds to *O. penicillatus* (Philippi, 1870) (Clade II) and GQ294363.1 (ID "As37") belongs to *O. meridionalis* (Clade V); in our results, the species identification is inverted (sequence GQ294370.1—Sample ID "As 60" and GQ294363.1—ID "As37") and these are highlighted in red in the tree (Figure 3).

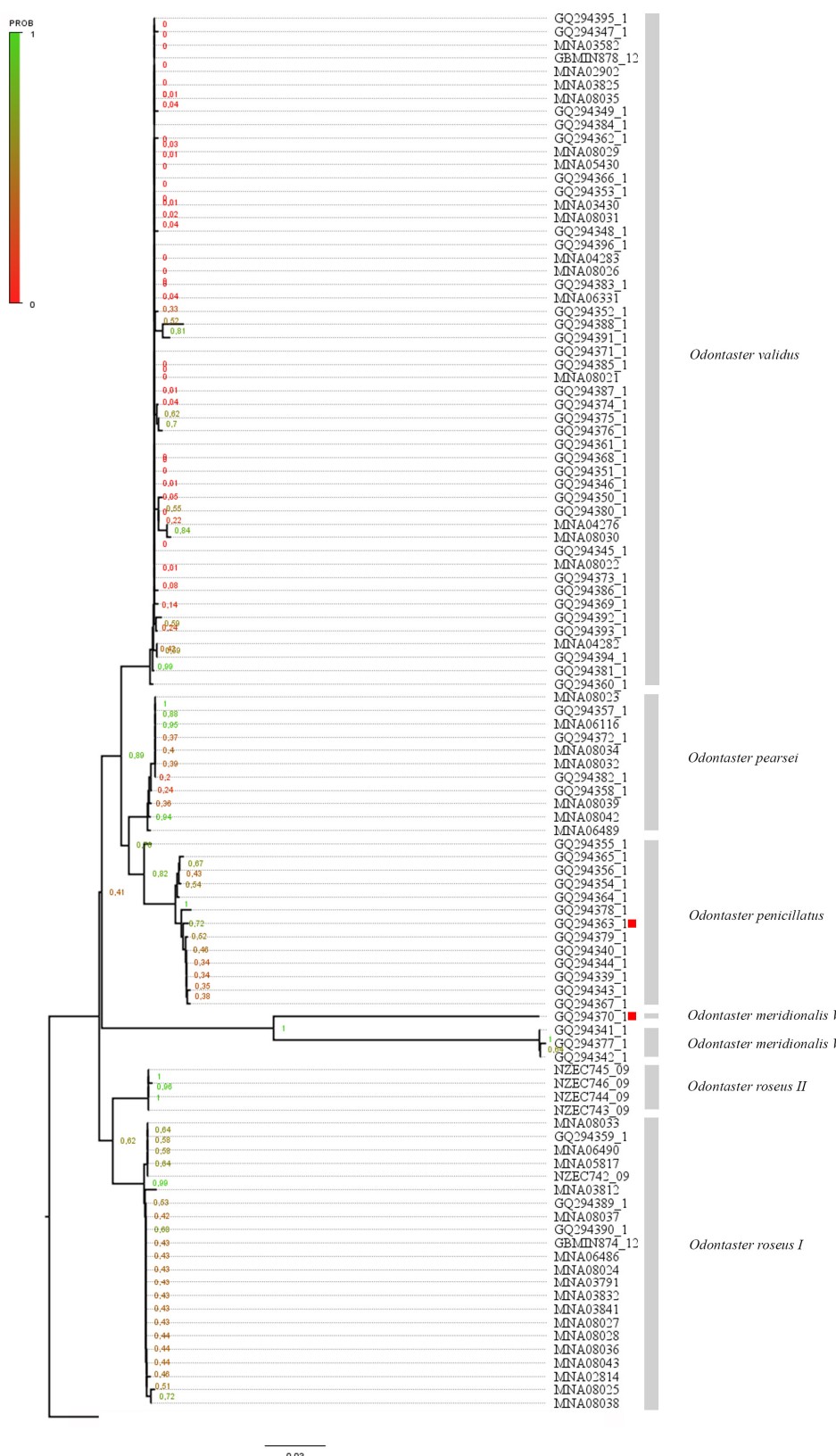

**Figure 3.** Maximum likelihood (ML) tree topology of the 105 COI sequences from the *Odontaster* genus. Posterior probability node values is shown on the tree with corresponding legend. Species names are reported near corresponding clades. Sequences GQ294370.1 and GQ294363.1 (from Janosik et al. [35]) are highlighted in red.

From the 65 sequences obtained from online data repository, 13 sequences of *Odontaster* from the Ross Sea area were retrieved (Table 4). Samples "As 33"," As 34", "As 69", "As 70", "As 71", and "As 72" from Janosik et al. [35] were not included in the analysis because the sequences were not available in a public database repository.

Overall, the analysis (Table 4) highlights many discrepancies between the sequence identification available in the online information systems and the results of our study. From BOLD, four samples from outside the Ross Sea (sample ID: 38186, 38512-1, 38719-1, and 38719-2) reported as *O. meridionalis* define a new clade with affinity for *O. roseus* (here denominated as *O. roseus II* for clarity). One sample from the Ross Sea was identified by us as *O. roseus* but is also incorrectly reported as *O. meridionalis* in BOLD (sample ID: 36438). Of the two *Odontaster* samples reported in Heimeier et al. [81] available from BOLD and Gen-Bank, one is correctly identified as *O. validus* (sample ID: A04N.08); the other one is labeled *O. meridionalis,* but we determined that it belongs to *O. roseus I* (sample ID: A02.15T). Our molecular results and current taxonomical identifications are reported in Supplementary File S3. Sequence identification of samples belonging to *O. meridionalis* (Clade V and VI) in Janosik et al. [35] remain under investigation. Moreau [7] findings suggested that the sequenced specimens might even belong to another family due to the large (COI) genetic distances involved. Such mismatches between morphological and molecular identifications, however, are a frequent outcome in DNA barcoding.

### 3.3. Morphological Analysis

Following the "reverse taxonomy" approach, morphological analyses were conducted for a re-examination of our molecular results on available specimens. The first feature we focused on was life coloration (Figure 4), using pictures of live specimens taken during expeditions. Organisms included in Clades I and II (corresponding to *O. roseus* and *O. pearsei*) presented a yellow or orange coloration. The yellow coloration of *O. roseus* in our samples differs to the original species description in Janosik and Halanych [34], in which the color, defined as rosy to drab red and tan, of their samples determined the choice of the descriptor "roseus" for the species name. Sample voucher MNA-08042 corresponds to a juvenile organism of *O. pearsei* and presented as pale-yellow coloration, which was slightly different from adults (Figure 4). Clade III, corresponding to *O. validus*, included two different colorations of morphotypes. Some specimens were characterized by the typical dark pink/red color (e.g., MNA-03825, Figure 4) and others had an orange coloration (e.g., MNA-02902, Figures 4 and 5). The co-occurrence in the same areas as species with the same coloration makes rapid identification very difficult, especially during diving or ROV sampling operations (Figure 5).

The second step of our morphological analysis focused on skeletal features, such as accessory structures and spines. We based our morphological analysis on the published descriptions and keys from Fisher [64] and Clark [56], with the addition of the unique characters highlighted by Janosik and Halanych [34], who suggest focusing on the number and length of paxillar spines, as well as differences in marginal plates and marginal spines to discriminate *O. roseus* and *O. pearsei* (Figure 6). The main morphological features used to identify species from the original description [34] are as follows:

*O. validus* Koehler, 1906: radial paxillae with about a dozen spinelets that are smooth, slender, and tapering; five actinal plate chevrons; actinal plates with up to seven similar, slender spinelets that are even from base to tip; two to three furrow spines.

*O. roseus* Janosik & Halanych, 2010: abactinal plates with distinct tabulum crowned with truncate paxillae, comprising 10–12 spinelets per plate; four complete actinal plate chevrons; actinal plates with spines of different lengths (8–10), specifically with one prominent spine in the middle.

*O. pearsei* Janosik & Halanych, 2010: abactinal plates with distinct tabulum crowned with truncate paxillae, comprising 16–20 spinelets per plate; three complete actinal plate chevrons; actinal plate with slender tapering (from tip to base) spines of equal length (5 to 8).

**Table 4.** A list of all the *Odontaster* sequences from the Ross Sea area available from online databases. Samples ID As 33, 34, 69, 70, 71, and 72 from Janosik et al. [35] are listed in the paper but sequences were not available in GenBank.

| Sample ID | Sequence Code BOLD | Sequence Code GenBank | Mined from | Wrong ID | Correct ID | Year | Location | BOLD BIN | Published |
|---|---|---|---|---|---|---|---|---|---|
| 36438 | NZEC742-09 | | BOLD | *O. meridionalis* | *O. roseus I* | 2008 | Ross Sea | BOLD:AAE2388 | |
| 38186 | NZEC743-09 | | BOLD | *O. meridionalis* | *O. roseus II* | 2008 | Out Ross | BOLD:AAE2389 | |
| 38512-1 | NZEC744-09 | | BOLD | *O. meridionalis* | *O. roseus II* | 2008 | Out Ross | BOLD:AAE2389 | |
| 38719-1 | NZEC745-09 | | BOLD | *O. meridionalis* | *O. roseus II* | 2008 | Out Ross | BOLD:AAE2389 | |
| 38719-2 | NZEC746-09 | | BOLD | *O. meridionalis* | *O. roseus II* | 2008 | Out Ross | BOLD:AAE2389 | |
| A02.15T | GBMIN874-12 | GU227088.1 | GenBank | *O. meridionalis* | *O. roseus I* | 2002 | McMurdo Sound | BOLD:AAE2388 | Heimeier et al., 2010 |
| A04N.08 | GBMIN878-12 | GU227092.1 | GenBank | | *O. validus* | 2004 | Cape Hallett | BOLD:AAK3286 | Heimeier et al., 2010 |
| As 68 | | GQ294374.1 | GenBank | | *O. validus* | 2011 | Ross Sea | | Janosick et al., 2011 |
| As 86 | | GQ294384.1 | GenBank | | *O. validus* | 2011 | Ross Sea | | Janosick et al., 2011 |
| As 87 | | GQ294385.1 | GenBank | | *O. validus* | 2011 | Ross Sea | | Janosick et al., 2011 |
| As 88 | | GQ294386.1 | GenBank | | *O. validus* | 2011 | Ross Sea | | Janosick et al., 2011 |
| As 33, 34, 69, 70, 71,72 | | | | | *O. validus* | 2011 | Ross Sea | | Janosick et al., 2011 |
| MNA-3582 | TCTNB082-15 | MK811555 | GenBank | | *O. validus* | 2019 | Terra Nova Bay | BOLD:AAK3286 | Rossi et al., 2019 |
| MNA-4276 | TCTNB079-15 | MK811610 | GenBank | | *O. validus* | 2019 | Terra Nova Bay | BOLD:AAK3286 | Rossi et al., 2019 |

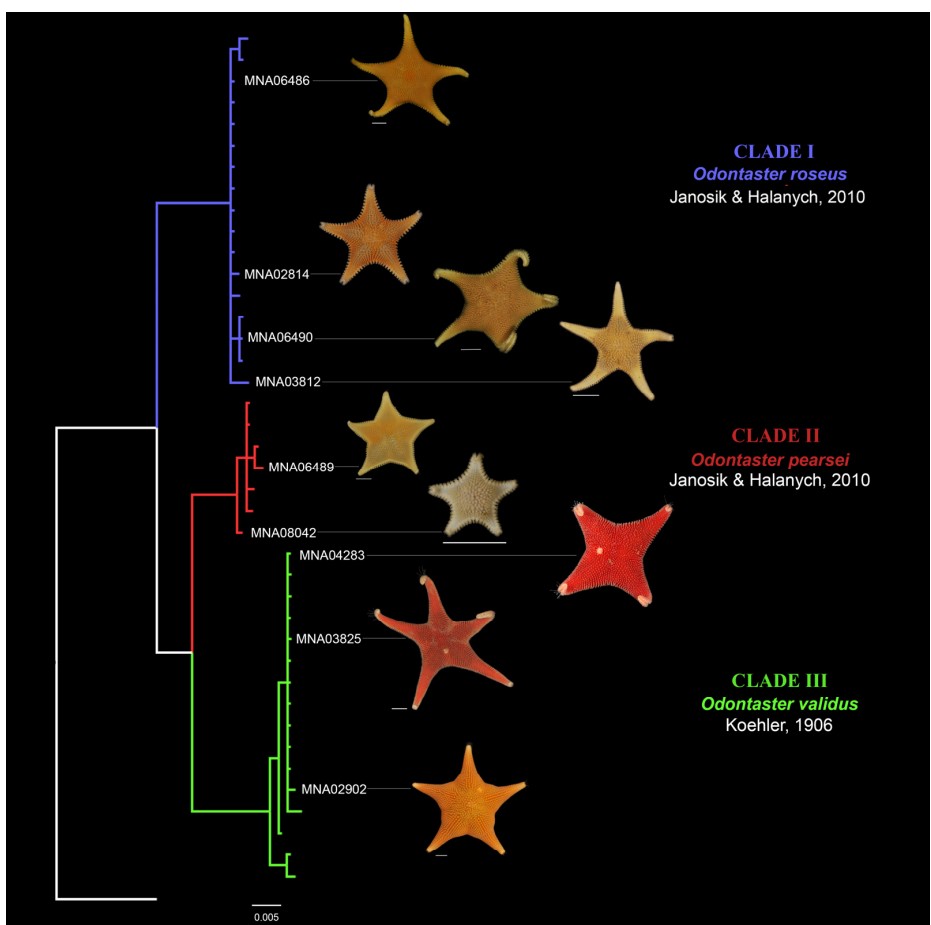

**Figure 4.** *Odontaster* morphology variability of selected specimens. In the tree highlighted in blue: *O. roseus* Janosik & Halanych, 2010, characterized by an orange coloration; red: *O. pearsei* Janosik & Halanych, 2010, characterized by an orange coloration; green: *O. validus* Koehler, 1906, with dark pink/red or orange coloration. Scale bar: 1 cm in grey.

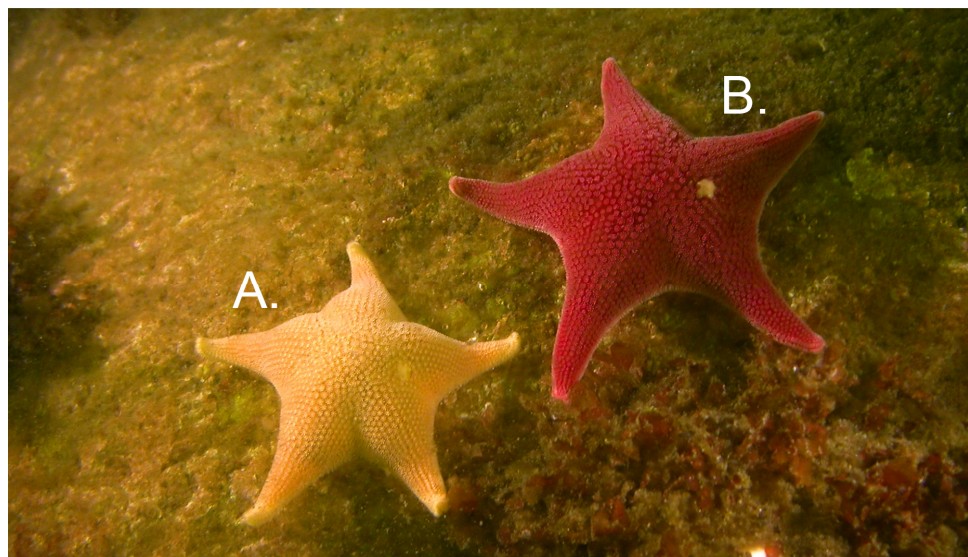

**Figure 5.** Two specimens of *Odontaster validus* (**A**,**B**) photographed by Stefano Schiaparelli during a dive in Road Bay (Terra Nova Bay area) at ~20 m depth. The orange yellow specimen ((**A**) in figure) corresponds to the sequenced MNA-02902 (also in Figure 4).

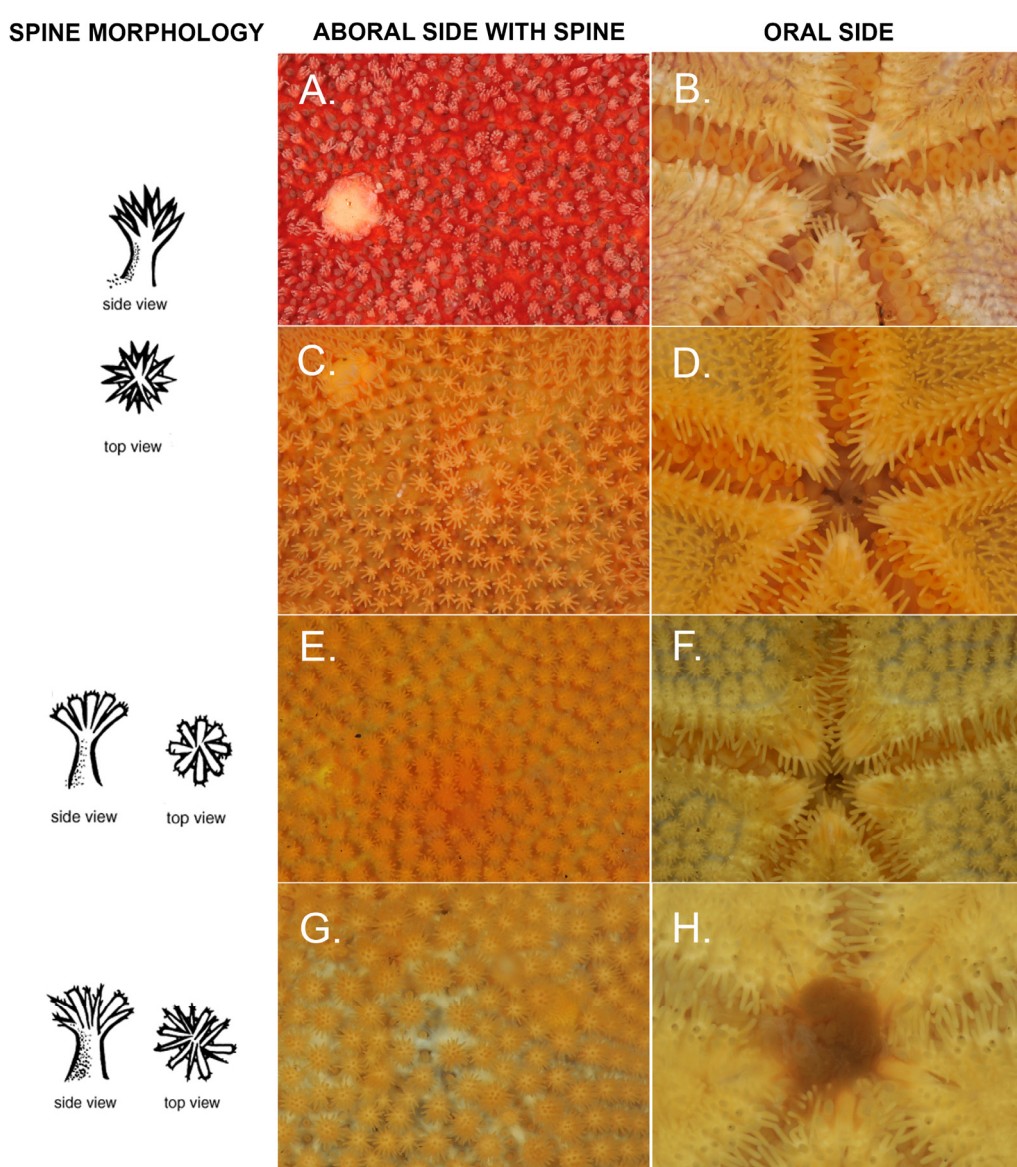

**Figure 6.** Photographic details of aboral side with spine on paxillae and oral close up, respectively: (**A**,**B**) *O. validus* red morphotype; (**C**,**D**) *O. validus* orange morphotype; (**E**,**F**) *O. roseus*; (**G**,**H**) *O. pearsei*. Drawings of peculiar spine morphology for each species from Janosik & Halanych [34].

The results of the external skeletal structures analysis of our samples were congruent with the species description and in agreement with the species partition resulting from the molecular analyses based on COI (Figure 2). This finding makes the occurrence in the TNB area of the three species robust. As suggested in a previous paper [34], the two species *O. roseus* and *O. pearsei*, reported for the first time in the Ross Sea with this work, should not be considered cryptic but merely unrecognized biodiversity that escaped identification until now.

*3.4. Scientific Literature Revision of Odontaster in the Ross Sea Quadrant*

We identified 93 articles that referred to the Ross Sea (Figure 7) that included 43 publications specifically mentioning the Terra Nova Bay area (Figure 8). All these papers were classified according to the main topic treated in the paper (Supplementary Files S4 and S5). We recognize that there is a possibility that some works, particularly those in the "grey literature", may not have been detected by the research methods we used for this article and, therefore, may not have been included in our review.

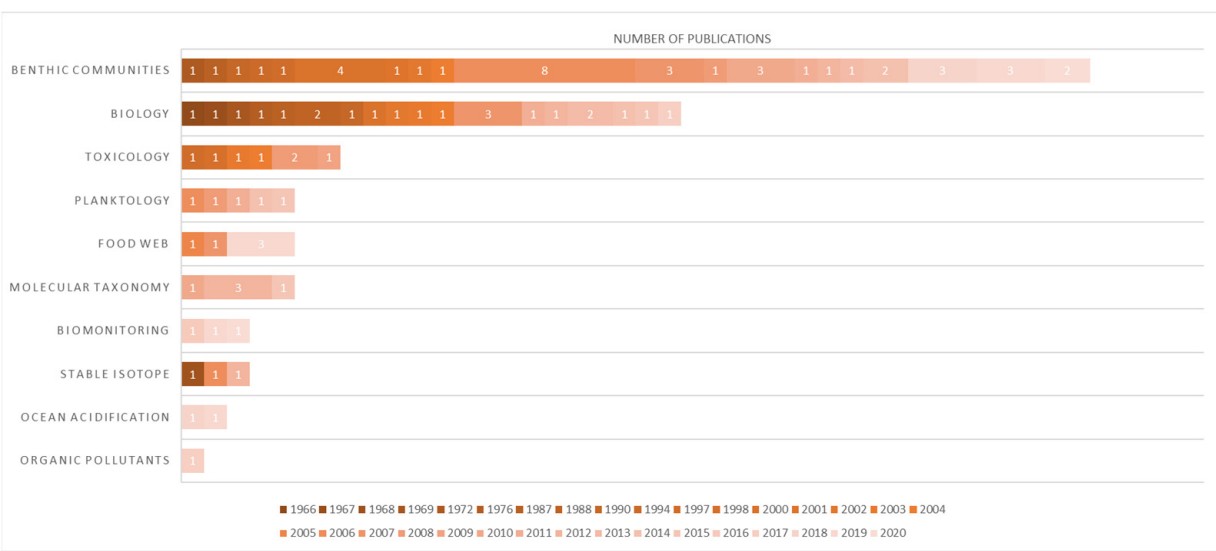

**Figure 7.** Graphical ranking representation of the 93 publications analyzed for this work for the Ross Sea. Each paper was classified into a general category. The publications are color-coded based on the year of publication (which runs from 1966 to 2020). The data refer to available literature in February 2022.

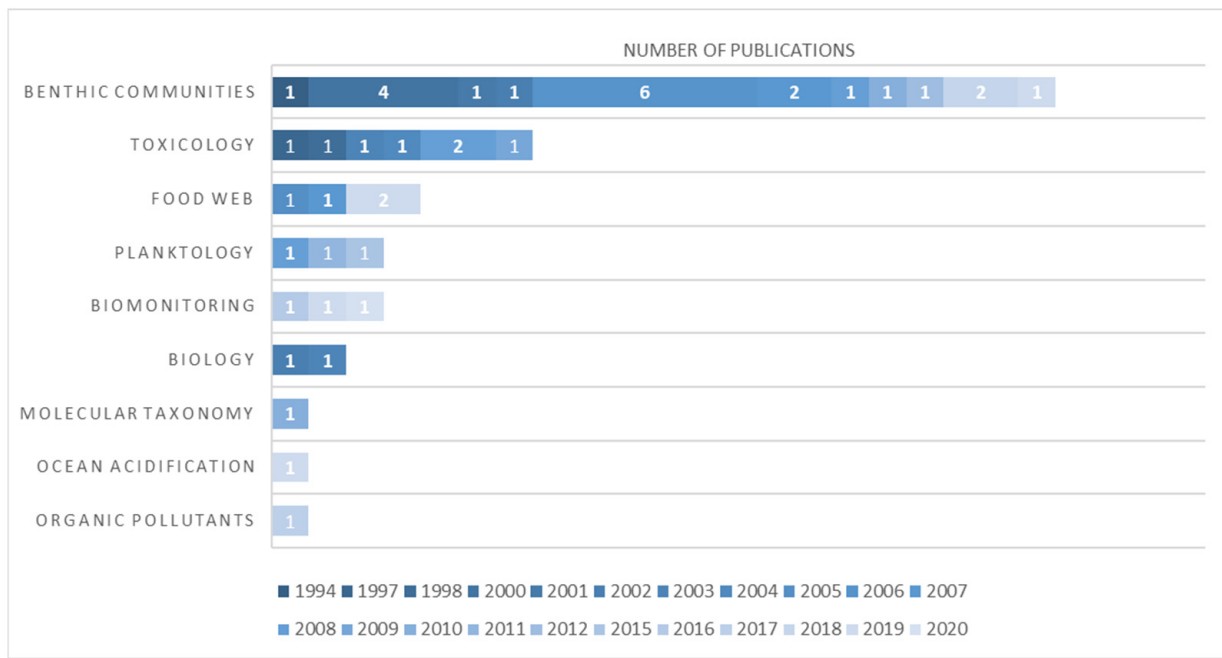

**Figure 8.** Graphical ranking representation of the 43 publications analyzed for this work in Terra Nova Bay area. Each paper was classified into a general category. The publications are color-coded based on the year of publication (which runs from 1972 to 2020). The data refer to available literature in February 2022.

Modern scientific investigations of the Ross Sea were initiated during the International Geophysical Year of 1957 and continue today. As a result, the Ross Sea is now one of the most intensively studied regions in the Southern Ocean. Ross Sea studies have greatly benefited from the presence of the McMurdo Station, located adjacent to McMurdo Sound on Ross Island. Research activities are also carried out by the Italian Mario Zucchelli research base, located in Terra Nova Bay 280 km to the north, and at the Scott base, managed

by New Zealand [82]. Many publications have followed over the years, and *Odontaster* has certainly represented a frequent subject of study confirmed by the 93 publications we found (Figure 7). In detail: 40 works were identified of benthic communities, 22 of biology, 5, respectively, of planktology, food webs, and molecular taxonomy. Minor contributions were found on biomonitoring and isotopes (3 each), ocean acidification (2), and organic pollutants (1). However, despite these numerous studies and various fields of research, the two lineages of *O. roseus* and *O. pearsei* went unnoticed until now. Most of the works conducted on *Odontaster* in the TNB area were, as expected, focused on the characterization of benthic communities (21), followed by toxicology (7) and food web (4), with the remaining being planktology and biomonitoring (3), biology (2), molecular taxonomy (1), ocean acidification (1), and organic pollutants (1) (Figure 8). Many scientific contributions highlight the important role which *O. validus* has in the sublittoral ecosystem (e.g., [54]) and the local abundance of this species in the area [42,83]. Only two papers out of the 43 analyzed applied a molecular approach to determine species identification. Heimeier et al. [81] used a combination of different markers (16 s, 18 s, and COI) to identify invertebrate larvae. They reported only the presence of *O. validus* in Cape Hallet and *O. meridionalis* in the Ross Sea. Our analysis of their *O. meridionalis* sequences, however, shows erroneous identification of that specimen (GenBank accession number: GU227088.1), which we found to belong to *O. roseus* (Table 4). The other work was from Rossi et al. (2019) [84], which focused on food web structure in TNB ecosystems. In this paper, COI sequences were used to crosscheck morphological determinations. Here, two *O. validus* (GenBank accession numbers: MK811555, MK811610) and one *Odontaster* sp. (MNA-04290, Annex 1) were reported, but the latter was not characterized at the molecular level. In more recent years, two other studies focusing on food web complexity in Terra Nova Bay were published by Signa et al. [43] and Caputi et al. [85]. In these papers, the species *O. meridionalis* and *O. validus* are considered as key players of benthic food webs, by being apex predators (e.g., Figure 4 in Caputi et al. [85]). Here, however, the lack of knowledge of true biodiversity in the area and the absence of molecular identifications led to the incorrect assumption that a "yellow *Odontaster*" is automatically an *O. meridionalis,* perpetuating the misidentifications of this species in the area. The lack of molecular data and/or museum vouchers for these specimens prevents correct determination of which one of the "yellow *Odontaster*" was involved.

## 4. Discussion

Although many studies have attempted to estimate biodiversity in the Southern Ocean, answering this question is not straightforward. In the present research, the biodiversity of the genus *Odontaster* in the Terra Nova Bay area (Ross Sea) was investigated in detail. Notably, our work has demonstrated that biodiversity knowledge could be considerably underestimated even in well-studied Antarctic areas and for iconic species. Although sea stars of the genus *Odontaster* are among the most frequently studied organisms in the Antarctic, two previously unrecognized species are reported for the first time from the Terra Nova Bay area (Ross Sea). This study complements the taxonomic and DNA barcoding effort of the Southern Ocean and highlights the necessity of revision even in the case of iconic and common organism.

There is considerable scientific literature reporting the presence of *O. validus* and *O. meridionalis* in the Ross Sea. However, the famous proverb "not all that glitters is gold" seems to describe very well the current situation, where "yellow *Odontaster*" were automatically assigned to *O. meridionalis*.

The new taxonomic evidence and the revision of public molecular databases showed several incorrect identifications for this genus in the literature. Especially in the shallow waters of Terra Nova Bay and McMurdo, where scientific activity has been intense, the presence of *O. validus* and *O. meridionalis* is widely reported, and they have been the subject of numerous scientific studies and experiments (Figures 7 and 8; Supplementary Files S4 and S5). The identification of these specimens was mostly undertaken using only

morphological traits, and the few molecular data show identification errors deriving from morphological recognition. Organism coloration was considered a sufficient trait for species recognition and it is possible that the existence of the well-known "McMurdo identification guide" (http://www.peterbrueggeman.com/nsf/fguide/echinodermata.pdf, accessed on 8 February 2022) [86], widely used in the field especially by research parties working the Ross Sea area, could perhaps represent a common source of these problems. On the other hand, other field guides report only these two species also for the Weddell quadrant of the Southern Ocean (e.g., [87]).

Thanks to the scientific contribution of Janosik & Halanych [34] on *Odontaster* from the Antarctic Peninsula, the existence of unrecognized biodiversity even in well-known areas and of iconic widely studied organisms has been brought to light. With our work, based on integrated molecular and morphological data, the presence of *O. validus* has been confirmed in TNB, and we report for the first time the species *O. roseus* and *O. pearsei*. These species, "as expected", were misidentified until very recently as *O. meridionalis*. In addition, we also report the existence of another "confounding factor", i.e., the presence of orange–yellow morphs of *O. validus*. The data presented here also demonstrate the existence of a yellow morphotype of *O. roseus* that differs from the rosy to drab red and tan coloration in the original species description. These three "yellow" sea star species live sympatrically and thus life coloration is a truly misleading character when "yellow morphs" have to be determined.

Correct identifications of *O. roseus* and *O. pearsei* can be easily achieved by using DNA barcoding and skeletal features, especially the number of spines on abactinal plates and spine length, as well as differences in marginal plates and marginal spines. Although in our case the use of morphological traits has made it possible to distinguish the species, particular caution should be employed when the identification of species depends on the morphological characteristics commonly proposed. As reported in the literature [64,88], different morphological features used to separate species of the genus *Odontaster* in Antarctica are highly variable and sufficiently variable to make them, at best, poor indicators of species-level differences in this genus.

Identification is, of course, possible for preserved specimens, whereas the determination of species in ROV images is simply not achievable. This highlights the irreplaceable role and resource of museums as biological specimen repositories and the relevance of their constant effort in curation of preserved specimens.

So far, based on our new data and on a thorough check of available COI sequence data available in GenBank, there is no molecular or morphological evidence to sustain the presence of *O. meridionalis* in the Ross Sea. However, the availability of molecular data for the area is still limited and further investigations, especially of offshore "yellow morphs", are necessary. Implementation of analysis of morphological traits and the increasing availability of molecular tools will improve identification of this species to be easier, faster, and more reliable in the future.

The revision of the morphological identification is not the only urgent action required to update the scientific information: with the review of the molecular data available online, we observed some incorrect classifications in BOLD and GenBank public databases that will need to be amended in the future. MOTUs correct taxonomic identification and the use of public sequence databases as exploration tools to evaluate taxonomic identification, the specificity, and robustness of the identification query (to species level or higher taxon) strongly depend on the related reference sequences available. The possibility of misleading identification carried out could have led to erroneous information flow into other science fields with inaccuracies that would persist in the scientific literature. A joint action of revision is fundamental for understanding the current level of diversity, speciation events of the past, and for implementing actions aimed at the conservation of these ecosystems and the species that occupy them. All this information is really important in the study area and in the future monitoring activities that are requested by the conservation measures of Annex 91-05/C [59] of the Ross Sea Marine Protected Area.

The current paper represents a further contribution of the Italian National Antarctic Museum (MNA)—Genoa section, as custodian of biodiversity data for the Ross Sea area. Many contributions to the Antarctic Biodiversity Portal have been published by MNA over the years, with the aim of increasing the knowledge of the area [89–96] (http://www.biodiversity.aq, accessed on 8 February 2022). It is desirable that in the next years, all available museum collections will be subject to molecular identifications in order to precisely determine species ranges and occurrences, key data for all monitoring activities.

**Supplementary Materials:** The following supporting information can be downloaded at: https://www.mdpi.com/article/10.3390/d14060457/s1. Supplementary Materials M1: COI sequence dataset produced in this study. Supplementary File S1: List of specimens analyzed in this study and corresponding GenBank accession numbers for COI sequences; File S2: Species partition methods; File S3: Tree topology of the 105 *Odontaster* analyzed and species partition methods; File S4: Literature review list for the Ross Sea area (the data refer to available literature in February 2022); File S5: Literature review list for the Terra Nova Bay area (the data refer to available literature in February 2022). The data presented in this study are openly available in GenBank (accession numbers: MK811555, MK811610, ON103472-ON103509).

**Author Contributions:** Data curation, A.G.; formal analysis, A.G.; funding acquisition, S.S.; investigation, A.G., M.C.A. and S.S.; resources, A.G., M.C.A., B.D., C.M. and S.S.; supervision, S.S.; validation, A.G.; visualization, A.G.; writing—original draft, A.G.; writing—review and editing, A.G., M.C.A., B.D., C.M. and S.S. All authors have read and agreed to the published version of the manuscript.

**Funding:** This research was funded by the project "BAMBi" (Barcoding of Antarctic Marine Diversity; PNRA 2010/A1.10; PI Stefano Schiaparelli) and "TNB-CODE" (Terra Nova Bay barCODing and mEtabarcoding of Antarctic organisms from marine and limno-terrestrial environments; PNRA 16_00120; PI Stefano Schiaparelli). The Italian National Antarctic Museum (MNA)—Genoa section played an essential cooperation role with the projects funded for biological specimen repository and outreach activity. Authors are grateful to the Italian National Antarctic Scientific Commission (CSNA) for the endorsement of this initiative and to the Italian National Antarctic Museum (MNA) for the financial support.

**Institutional Review Board Statement:** All the sampling activities in Antarctica were authorized by the Italian National Antarctic Program (PNRA).

**Informed Consent Statement:** Not applicable.

**Data Availability Statement:** In accordance with FAIR principles, the COI sequence dataset produced in this study (Supplementary Materials M1) could be found in the Supplemental Materials and openly available in GenBank (https://www.ncbi.nlm.nih.gov/genbank/, accessed on 8 February 2022, accession numbers: MK811555, MK811610, ON103472-ON103509).

**Acknowledgments:** This paper is an Italian contribution to the CCAMLR CONSERVATION MEASURE 91-05 (2016) for the Ross Sea region Marine Protected Area, specifically addressing the priorities of Annex 91-05/C. Many thanks are due to D.K.A. Barnes and J. Murray who contributed to significant linguistic improvements. We are grateful to W. I. Ausich and to two anonymous reviewers, whose comments and suggestions were of great help in improving the quality of this paper.

**Conflicts of Interest:** The authors declare no conflict of interest. The funders had no role in the design of the study; in the collection, analyses, or interpretation of data; in the writing of the manuscript, or in the decision to publish the results.

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
