# Peer review of "Not All That Glitters Is Gold: Barcoding Effort Reveals Taxonomic Incongruences in Iconic Ross Sea Sea Stars"

_diversity, doi:10.3390/d14060457_

Round 1
Reviewer 1 Report
Please see attached file for comments.

Author Response
Dear Reviewer,
thank you for your time and effort in reviewing our ms. Your comments have been taken in great consideration and changes you suggested have been made to improve the ms.
Please see the attachment.
With all my best regards,
Alice Guzzi

Reviewer 2 Report
Dear Editor,
The manuscript "Not all that glitters is gold: barcoding effort reveals taxonomic incongruences in iconic Ross Sea seastar" provides relevant, innovative data and raises an important question about the diversity of Odontaster in Antarctica. However, my main criticism of the manuscript is the attribution of the identification mistakes of the species of the genus solely to morphology/taxonomy. Also, the authors also place a lot of weight on color. However, color is not a diagnostic character, it just be an indication to a single species when the local diversity is already known. Therefore, I recommend that the authors soften their criticisms and not directly relate the identification mistakes (which are often not made by specialists) to the taxonomy/morphology of the group. Finally, it is necessary to review some small issues of text formatting.
In view of the above, I recommend publishing the manuscript with minor modifications.
Author Response

(The authors gave the same response as above.)

Reviewer 3 Report
This is an excellent paper. I suggest that the authors stress even more that these results demonstrate that both bar coding AND morphological study are needed in our continuing effort to document Earth's biodiversity. This is clearly the case in the waters around Antarctica, and I am this should be done in most habitats globally. Either approach is lacking in the absence of the other.
I recommend this paper for publication with moderate revision to the English wording
Assuming that the numbers and letters on Figures 2 and 3 are important (because they are there), most are much too small to read.
Throughout the middle of the manuscript (most of the ms) the italics font for genus and species names was dropped. Please correct.
The enclosed document is a marked pdf. This includes a number of corrections to the text and many suggestions for improving clarity of the writing. Please consider these remarks while revising this manuscript.

Author Response

(The authors gave the same response as above.)

Round 2
Reviewer 1 Report
Thank you for the edited manuscript. The authors have changed a lot, but not all I asked for (including increasing the numbers of dna sequences in order to compare with). The manuscript still lacks a good and logic structure, and I recommend rejecting this ms.